# Moderate Prenatal Alcohol Exposure Increases Toll-like Receptor Activity in Umbilical Cord Blood at Birth: A Pilot Study

**DOI:** 10.3390/ijms25137019

**Published:** 2024-06-27

**Authors:** Jessie R. Maxwell, Shahani Noor, Nathaniel Pavlik, Dominique E. Rodriguez, Lidia Enriquez Marquez, Jared DiDomenico, Sarah J. Blossom, Ludmila N. Bakhireva

**Affiliations:** 1Department of Pediatrics, University of New Mexico, Albuquerque, NM 87131, USA; 2Department of Neurosciences, University of New Mexico, Albuquerque, NM 87131, USA; 3College of Pharmacy, University of New Mexico, Albuquerque, NM 87131, USA; 4Substance Use Research and Education (SURE) Center, College of Pharmacy, University of New Mexico, Albuquerque, NM 87131, USA

**Keywords:** prenatal alcohol exposure, in utero exposure, toll-like receptor agonists, cytokine levels, immunological development, fetal alcohol spectrum disorders, umbilical cord blood

## Abstract

The prevalence of prenatal alcohol exposure (PAE) is increasing, with evidence suggesting that PAE is linked to an increased risk of infections. PAE is hypothesized to affect the innate immune system, which identifies pathogens through pattern recognition receptors, of which toll-like receptors (TLRs) are key components. We hypothesized that light-to-moderate PAE would impair immune responses, as measured by a heightened response in cytokine levels following TLR stimulation. Umbilical cord samples (10 controls and 8 PAE) from a subset of the Ethanol, Neurodevelopment, Infant and Child Health Study-2 cohort were included. Peripheral blood mononuclear cells (PMBCs) were stimulated with one agonist (TLR2, TLR3, TLR4, or TLR9). TLR2 agonist stimulation significantly increased pro-inflammatory interleukin-1-beta in the PAE group after 24 h. Pro- and anti-inflammatory cytokines were increased following stimulation with the TLR2 agonists. Stimulation with TLR3 or TLR9 agonists displayed minimal impact overall, but there were significant increases in the percent change of the control compared to PAE after 24 h. The results of this pilot investigation support further work into the impact on TLR2 and TLR4 response following PAE to delineate if alterations in levels of pro- and anti-inflammatory cytokines have clinical significance that could be used in patient management and/or attention to follow-up.

## 1. Introduction

Prenatal alcohol exposure (PAE) remains a common occurrence across the world, including in the United States, with as many as 12% of pregnant women consuming alcohol [1,2,3]. Alcohol readily crosses the placenta, and thus can directly impact fetal development [4,5]. The consumption of alcohol during pregnancy remains a leading cause of birth defects and neurodevelopmental disorders in the United States. Fetal alcohol spectrum disorders (FASDs) is the umbrella term used to describe multiple phenotypes that can occur following PAE [1,2,6,7]. Many children with FASDs have lifelong disabilities, including cognitive deficits, motor functioning delays, and impaired behavioral and adapting skills [2,6,7,8,9]. Children with fetal alcohol syndrome (FAS), the most severe phenotype under FASDs, were reported to have impaired immunologic functioning resulting in an increased risk of pneumonia, meningitis, and sepsis compared to the age-matched controls [10]. Subsequent studies have found that infants with PAE have an increased risk of infection [11,12,13]. Infants with PAE who are born small for their gestational age might be particularly at risk: a 2.5-fold increased risk of infection following any alcohol exposure and a 3–4-fold increase among those exposed to more than seven drinks per week [13]. Investigations in women consuming low-to-moderate alcohol during pregnancy and any resultant impact on fetal immune development are less studied.

Heavy or binge drinking during pregnancy increases the risk of prematurity, which, in turn, increases the risk of respiratory virus or bacterial infection in newborns [14]. The findings of increased infection risk following PAE continue to be replicated in both human studies as well as in pre-clinical models, with impairments in adaptive responses including cell-mediated and humoral immunity. Collectively, these immune impairments not only increase the risk of infections but also heighten the risk of certain malignancies [14,15,16,17,18,19]. While the reason for these associations is not clear, it is currently thought that PAE alters the developmental trajectory of the immune system beginning during fetal development and the early neonatal period when the innate immune system is more functional. The mechanisms of PAE’s impact on the infant’s immune function are thought to occur through a variety of effects. One example is evidenced by impaired macrophage maturation and function [12,14]. The innate immune system is the first to develop in the fetus, with the adaptive immune system developing later. As it is difficult to study immunological parameters in infants, assessing the production of innate-specific immune mediators from umbilical cord blood can provide important clues early on that may help predict immune system function and adverse outcomes later in life, such as possible susceptibility to cancer or infection.

In preclinical studies, prenatal chronic heavy alcohol use is associated with increased pro-inflammatory cytokine levels, interleukin-1-beta (IL-1β), interleukin-6 (IL-6), and tumor necrosis factor alpha (TNF-α) in both the mother and the fetus. Additionally, when cord blood-derived peripheral blood mononuclear cells (PBMCs) from drug-free individuals were isolated and stimulated in vitro with lipopolysaccharide (LPS) and various concentrations of alcohol, a blunted effect of alcohol on LPS-induced cytokine secretion was observed [20]. While the relationship between PAE and the developing immune system continues to be investigated, many studies have focused on preclinical models or infant outcomes after birth. Recent studies demonstrated that the adult offspring of rats who consumed moderate levels of alcohol during pregnancy had exaggerated levels of pro-inflammatory cytokines, including IL-1β, IL-6, and TNF-α, in the brain and peripheral immune system following stimulation with TLR4 agonist—LPS [21,22]. While these studies shed light on the immediate and long-term effects of moderate-to-high fetal alcohol exposure, whether low-to-moderate levels of PAE can affect fetal innate immune activity in humans is unknown. To investigate this association in the current study, we sought to determine if alterations in cytokine levels could be measured at the time of delivery in infants with PAE and controls. Biomarkers of the fetal phase of immune development measured in umbilical cord blood offer great potential for improved diagnosis and prognosis value in FASDs and other adverse infant outcomes associated with PAE. We hypothesized that PAE would result in TLR-mediated dysregulated cytokine production in umbilical cord blood-derived leukocytes.

Cytokine production is important to maintain harmony within the body, with many conditions displaying abnormal production of anti-inflammatory cytokines, pro-inflammatory cytokines, or both. Alzheimer’s Disease continues to have advancements in the understanding of poor anti-inflammatory responses that lead to the induction of pro-inflammatory signaling and ultimately cause neuronal cell death [23,24,25]. Pro- and anti-inflammatory cytokine alterations are also thought to have a role in recurrent pregnancy loss [26,27]. Maintaining a balance of pro- and anti-inflammatory cytokines is critical for normal function. Thus, both pro- and anti-inflammatory cytokines are measured in this pilot study.

TLRs are pattern recognition receptors expressed in multiple types of innate immune cells, such as monocytes, dendritic cells, and macrophages. TLRs play a vital role in the modulation of the first line of innate immune defense [28]. Different TLRs recognize distinct or overlapping microbe-specific molecular signatures, known as pathogen-associated molecular patterns (PAMPs) that activate downstream signaling pathways. This results in the release of inflammatory mediators that generate immediate host defensive responses as well as the modulation of antigen-specific adaptive immunity [29,30]. While there is prior evidence from human fetal lymphocytes in vivo and in vitro studies and a growing body of preclinical data focusing on alcohol and potential TLR4 interactions [31], the effects of PAE on functional responses on other TLRs are relatively unknown. Thus, multiple TLR agonists (specifically those known to have immunostimulatory effects) were utilized to better characterize PAE-induced alterations, examining a more comprehensive profile of pro- and anti-inflammatory cytokines. This pilot study is meant to inform a larger-scale trial with a particular focus on those areas identified as most impacted in this smaller sample size.

## 2. Results

No significant differences between the study groups were observed in sociodemographic and medical characteristics except for some differences in marital status (Table 1). The level of alcohol exposure was light-to-moderate in the PAE group with the average AA/day across pregnancy [SD] = 0.45 [0.19], which is equivalent to approximately three drinks per week (Table 1).

### 2.1. Baseline Cytokine Levels

There were no significant differences in the baseline level of any cytokines prior to stimulation with a TLR agonist between the study groups (all *p*’s > 0.05, Figure 1, Figure 2, Figure 3 and Figure 4).

### 2.2. TLR4 Agonist Stimulation

In assessing the levels of pro-inflammatory cytokines 24 h after stimulation with LPS, no significant differences were observed between the PAE and control groups (F(1,2) = 0.00–1.52; *p* = 0.17–1.00 for all cytokines). The time comparison of cytokine levels following TLR4 stimulation was significant only from baseline measurements to 24 h (F(2,2) = 4.90–22.87; *p* < 0.05, *p* < 0.01, *p* < 0.001 for all cytokines over time).

Multiple pro-inflammatory cytokines had increased levels 24 h after stimulation with LPS compared to baseline in both the control and PAE groups, including IFN-γ (45% increase in controls vs. 49% in PAE, Figure 5), IL-12p70 (49% increase in controls vs. 57% in PAE, Figure 5), IL-1β (48% increase in controls vs. 71% in PAE, Figure 5), IL-2 (52% increase in controls vs. 66% in PAE, Figure 5), IL-6 (78% increase in controls vs. 86% in PAE, Figure 5), IL-8 (67% increase in controls vs. 72% in PAE, Figure 5), and TNF-α (69% increase in controls vs. 83% in PAE, Figure 1; all *p* < 0.05, Figure 1 and Figure 5). Additionally, after 24 h of stimulation with LPS, both groups had a significant increase in the levels of anti-inflammatory cytokines IL-4 (58% increase in controls vs. 66% in PAE, Figure 5) and IL-13 (59% increase in controls vs. 74% in PAE, Figure 5; all *p*’s < 0.05, Figure 1). The findings of increased pro-inflammatory cytokine levels after stimulation with LPS are not surprising, given the preclinical literature showing that PAE can increase cytokine levels that are not observed in TLR4-deficient mice [32,33,34]. Only the PAE group had a significant increase in the 24 h after stimulation with LPS compared to the baseline in levels of the anti-inflammatory cytokine IL-10 (*p* < 0.001, Figure 1B).

### 2.3. TLR2 Agonist Stimulation

PAM3CSK4, the TLR2 agonist, did not result in significant differences between the control and PAE groups. However, TLR2 stimulation did significantly increase the levels of multiple pro-inflammatory cytokines from baseline to 24 h within groups, including IFN-γ (45% increase in controls vs. 56% in PAE, Figure 5), IL-12p70 (46% increase in controls vs. 62% in PAE, Figure 5), IL-2 (57% increase in controls vs. 68% in PAE, Figure 5), IL-8 (78% increase in controls vs. 72% in PAE, Figure 5), and TNF-α (69% increase in controls vs. 84% in PAE, Figure 5) in both groups (all *p*’s < 0.05, Figure 2). There was a significant increase only in the level of IL-1β in the PAE group (*p* < 0.01, Figure 2E) following TLR2 stimulation.TLR2 is embedded in the cell membrane, similar to TLR4, and activation can lead to the release of inflammatory cytokines, specifically IL-1β [35], as observed in this study. The levels of anti-inflammatory cytokines were also significantly increased in both groups, including increased levels of IL-13 (62% increase in controls vs. 75% in PAE, Figure 5) and IL-4 (46% increase in controls vs. 71% in PAE, Figure 5; all *p*’s < 0.05, Figure 2). Only the PAE group had increased levels of the anti-inflammatory cytokine IL-10 (*p* < 0.01, Figure 2B) following TLR2 stimulation.

### 2.4. TLR3 Agonist Stimulation

The stimulation with the TLR3 agonist, Poly I:C, resulted in a significant difference between the control group and the PAE group in the levels of pro-inflammatory cytokine IL-12p70 (11% increase in the control group vs. 39% in the PAE group, Figure 5, *p* < 0.05) and the anti-inflammatory cytokine IL-4 (10% increase in the control group vs. 53% in the PAE group, Figure 5, *p* < 0.05; Figure 3B and 3C, respectively). Additionally, the PAE group had a significant change in the level of IFN-γ from baseline to 24 h (23% increase in the control group vs. 43% in the PAE group, Figure 5, *p* < 0.05, Figure 3A). Poly I:C is a viral mimetic, and long-lasting immunomodulatory effects have been observed with a Poly I:C challenge following PAE [36], suggesting that these alterations could persist into adulthood. Levels of the remaining pro-inflammatory cytokines, including IL-1β, IL-2, IL-8, and TNF-α, were not significantly different 24 h after stimulation with Poly I:C.

Levels of the remaining anti-inflammatory cytokines following stimulation with the TLR3 agonist resulted in no significant changes in IL-10 or IL-13 levels.

### 2.5. TLR9 Agonist Stimulation

Stimulation with the TLR9 agonist, CpG ODN, did not result in any significant differences in anti-inflammatory cytokine level in either the percent change or the baseline to 24-h measurements in the control or PAE groups. The levels of pro-inflammatory cytokines IL-6 had a significant difference in the percent change between the control and PAE groups (5% increase in the control group vs. 40% in the PAE group, Figure 5, *p* < 0.05; Figure 4), with the PAE group having a higher percent change. The remaining pro-inflammatory cytokine levels did not significantly differ.

## 3. Discussion

The results of this study indicate that stimulation with TLR agonists resulted in variable levels of pro-inflammatory and anti-inflammatory cytokines in the PAE group compared to the control groups. While this is a pilot study, it is interesting to note that the utilization of this sensitive assay revealed differences in cytokine levels with the mild-to-moderate exposure levels—a novel finding that supports a larger-scale trial. Both pro- and anti-inflammatory cytokine levels were impacted, which may indicate an imbalance in specific pathways following PAE. Prior studies have revealed differences following chronic levels of heavy prenatal alcohol exposure, with moderate levels of exposure having heterogeneous results [16,20,21].

Differences in the rate of cytokine levels during the time course (between baseline, 3 h, and 24 h) following stimulation with the TLR4 agonist were also noted between exposed and unexposed groups. There was a significant difference in levels over time of IL-2, IL-4, and IL-6 between the groups over these three time points, suggesting a different progression of immune cytokine release in the presence of PAE. Taken together, this may indicate an alteration in the time response of the cells following PAE. Indeed, a study of cytokine levels in adults showed different cytokine levels at baseline and 6 h after the ingestion of alcohol with an increase in IL-8 [37]. The TLR4/myeloid differentiation primary response protein 88 (MyD88) pathway leads to the activation of nuclear factor-kB (NF-kB), which results in the release of IL-1β, IL-6, and TNF-α [38,39]. Binge drinking in adults has shown the suppression of the TLR4-MyD88 responses [40]. While there were no significant differences in the levels of IL-1β, IL-6, and TNF-α between the PAE and unexposed groups in our study, the PAE group had consistently lower levels of these cytokines.

Interestingly, in a study of children born preterm with periventricular leukomalacia-induced cerebral palsy, a disorder that impacts movement, altered inflammatory responses in PBMCs stimulated with LPS were found compared to control preterm infants with normal neurodevelopment [41,42]. Thus, in children with PAE, alterations in cytokine responses could contribute to abnormalities observed in neurodevelopment. Future studies should focus on assessing neurodevelopmental outcomes after measuring cytokine responses in infancy to determine if any correlation could be identified.

The PAE group had a significant increase in IL-1β levels following stimulation with the TLR2 agonist, while the control group did not have this change in level. This may indicate that PAE-related adaptations induce specific vulnerability to the TLR2 pathways, thus leading to increased IL-1β production. IL-1β is a potent pro-inflammatory cytokine and a key driver underlying impaired outcomes associated with autoinflammatory disorders and neonatal sepsis [43,44,45]. Interestingly, the Hepatitis C Virus is known to be recognized by the host through the TLR2 pathway, with resultant increased secretion of IL-10 and TNF-α, decreased IL-6 production, and augmented hepatic necroinflammatory activity [46,47,48]. The Hepatitis C Virus has known congenital transmission, and Hepatitis C Virus-positive individuals are significantly more likely to have a history of alcohol use compared to individuals without Hepatitis C Virus infection [49]. While the differences in IL-10 and TNF-α levels between the PAE and control groups were not significant in our study, it is interesting to note that the PAE group had higher levels of TNF-α on both experimental days and higher levels of IL-10 on one experimental day. The course of hepatitis C viral infection should be monitored very closely in those with known PAE, as this could pose a risk of exacerbating the disease. It would also be interesting to determine if cytokine levels are correlated with the likelihood of congenital Hepatitis C Virus transmission in those with PAE.

Interestingly, for TLR3 stimulation, the pro-inflammatory mediator production was not evident in the alcohol-exposed group. TLR3 not only induces inflammation, but TLR3 activity also detects viral particles (double-stranded RNA) and provides protective immunity against viral infections [50,51]. A preclinical study found that PAE resulted in increased and sustained pulmonary viral titers following influenza A infection [36]. While we cannot exclude the possibility that PBMCs from the PAE group may respond to TLR3 agonist at earlier or later time points than the 24 h post-stimulation, the differential sensitivity of circulatory immune cells to TLR3 agonist, observed in our study, may confer susceptibility to viral infections in children with FASDs.

The imbalance of pro- and anti-inflammatory cytokine levels, as noted in this pilot study, raises focus areas for future larger-scale trials. Specifically, TLR2 and TLR4 pathways seem the most impacted by PAE in this sample. The ubiquitous nature of TLR2 roles and functions could result in various clinical findings in those with alterations in that pathway, such as more common infections with gram-positive organisms [29,48]. Following a larger population after obtaining the cord blood sample could be one approach to determine if clinical differences are observed. Additionally, an improved understanding of the imbalance of pro- and anti-inflammatory cytokine levels may allow for treatment modalities to be identified and tested.

One strength of this study was the utilization of multiple TLR agonists. Commonly, the TLR4 agonist, lipopolysaccharide, has been used to determine the impact on cytokine levels in PAE [20,21]. This is likely due to the fact that TLR4 is one of the most studied TLRs to date [52]. We were able to expand the TLR agonists studied to better characterize the potential impact PAE has on immune response in the setting of various exposures, including bacteria and viruses. Additionally, this cohort has neurodevelopmental follow-up in the children that is currently underway. This provides a unique opportunity to begin investigating if the specific cytokine profiles observed at birth are associated with any identifiable neurobehavioral alterations later in life. This information would help inform whether altered patterns of cytokine levels detectable at birth could be prognostic for infants at high risk of altered neurodevelopment following prenatal alcohol exposure.

Limitations of this study include the small sample size available for this analysis, which resulted in the inability to examine the effects of co-exposures and other confounders. However, the sample did not contain patients with autoimmune disorders. As this study is observational, causation cannot be determined. The cell numbers plated in each well for TLR agonist stimulation on each assay day were chosen based on the cell yields across the samples being used with the goal of optimizing the ability to assay all experimental conditions across all samples. Therefore, the experimental day did influence the cytokine levels obtained. While this may be due to the difference in the number of cells plated, the overall patterns of cytokine levels were consistent between the experimental days. Additionally, while study participants had state-of-the-art measures of alcohol exposure during pregnancy, there is some variability in the pattern of alcohol use among participants, which may have contributed to some of the variability in the cytokine levels following TLR agonist stimulation. Those included in this pilot study were based on self-report only, without confirmation from the biomarkers of the parent study. This could introduce bias into the results observed, although the combination of prospective repeated TLFB interviews and targeted questions about binge drinking does result in a reasonable exposure [53]. The statistical transformations alleviated the severity of the violations noted in statistical analysis, but some deviations from model assumptions persisted. With additional samples, analyses could be conducted to look more specifically at the patterns of alcohol exposure and the resultant cytokine profile in the newborn. Future studies should focus on expanding the time course and doses utilized to deepen the knowledge from this pilot study.

## 4. Material and Methods

### 4.1. Ethanol, Neurodevelopment, Infant and Child Health Study-2 Cohort

A subset of the Ethanol, Neurodevelopment, Infant and Child Health Study-2 (ENRICH-2) prospective cohort was utilized for this study. The study design incorporated maternal structured interviews and the collection of umbilical cord blood samples at delivery. Women were grouped into PAE and control groups based on interview findings and ethanol biomarkers from biological specimens. Participants provided written informed consent, and the research protocol was approved by the University of New Mexico Health Sciences Center Institutional Review Board.

Initial study inclusion required participants to be 18 years of age or older, at least 12 weeks gestational age, residing in Albuquerque, New Mexico, and have a singleton pregnancy. All participants were screened to exclude confounding factors including severe maternal mental health disorders, severe fetal abnormalities, and use of methamphetamines and/or opioids. Co-exposures to nicotine and marijuana were allowed in both control and PAE groups and measured via self-report and urine drug screening both at enrollment and delivery. U.S. Drug Testing Laboratories Inc. analyzed the maternal urine samples in a “panel-07” drug screen to assess the use of amphetamines, barbiturates, benzodiazepines, cannabinoids, PCP, cocaine, and opiates. All participants (regardless of the disclosed amount of alcohol use) received counseling about the risks associated with alcohol use in pregnancy and were provided with resources, including a counseling hotline.

Alcohol use in the periconceptional period and during pregnancy was first ascertained via self-report in four Timeline Follow-Back (TLFB) interviews [53]. As this study focuses on samples obtained at delivery, the remaining postpartum TLFB interviews obtained for the ENRICH-2 cohort study are not applicable to this pilot study. The TLFB interview, completed by trained study staff, collected information from participants about alcohol consumption during the preceding 4 weeks, with one “standard drink” or standard drink unit (SDU) referring to any drink containing 14 g or approximately 0.6 fluid ounces of “pure” ethanol. The standard drink is equivalent to one 12-ounce can/bottle of regular beer, one 5-ounce glass of wine, 1.5 ounces of hard liquor, or one mixed drink [54]. This information (quantity and frequency) was used to calculate the ounces of absolute alcohol per day (AAD) and absolute ounces of alcohol per drinking day (AADD). AAD and AADD were both calculated to capture the average alcohol consumption as well as binge-like behavior, which are both important factors for the potential impact of prenatal alcohol exposure [55,56]. Two standard drinks equal approximately one ounce of absolute alcohol (AA) [55]. The TLFB approach is considered a “gold standard” for ascertaining PAE and was reported to be the most predictive of future neurodevelopmental outcomes [57].

Participants in the control group were required to demonstrate no more than minimal self-reported alcohol use during the periconceptional period: <13 standard drink units per month (<6.5 AA/month) and no binge drinking episodes (defined as 4 or more SDU on one occasion or 2 or more AA on one occasion) in the month surrounding the last menstrual period (LMP). To remain eligible as controls, participants were required to have no alcohol exposure during pregnancy. Participants were provisionally assigned to the PAE group based on reported perinatal alcohol consumption of ≥2 binge drinking episodes or ≥13 SDU in the periconceptional period—the month surrounding the LMP. Participants who reported greater than 13 SDU on the first prenatal TLFB or reported at least one binge episode anytime during pregnancy continued to be eligible for the PAE group. Alcohol exposure levels required in the PAE group follow greater than “minimal risk” exposure levels recommended for the diagnosis of PAE-related disorders and correspond to low-to-moderate exposure [58]. Typically, low exposure is defined as fewer than 3 drinks per week and moderate exposure is defined as fewer than 7 drinks per week [59,60]. The definitions used to classify participants into the PAE group are consistent with the Diagnostic and Statistical Manual of Mental Disorders, Fifth Edition for Neurobehavioral Disorder Associated with Prenatal Alcohol Exposure (ND-PAE), as well as those used in the parent ENRICH study [61,62,63,64,65,66]. Individuals included in this pilot study were selected based on self-reported alcohol intake, prior to knowing all biomarker results.

Pregnant participants who remained eligible for their respective study groups were followed to delivery. Following delivery, placenta and umbilical cord blood (4 mL of whole blood) samples were obtained for the study. Autoimmune disorder information was collected via self-report, with no individuals reporting any autoimmune disorders. In this pilot study to examine feasibility, samples from 18 infants (8 in the PAE and 10 in the control groups) were included in this analysis. Umbilical arterial cord blood was collected in EDTA-coated tubes; peripheral blood mononuclear cells (PBMCs) were isolated from these samples and processed within 24 h of delivery (see below). The utilization of arterial cord blood was chosen as this represents the infant’s blood and thus the PBMCs and responses are reflective of the infant.

Variables collected on the pregnant participants included maternal age, total number of years of education completed, highest educational level achieved, marital status, race, type of delivery, AAD, and AADD. Infant variables collected included gestational age at delivery, birthweight, and sex (see Table 1).

### 4.2. Peripheral Blood Mononuclear Cell (PBMC) Isolation and Stimulation In Vitro

PBMCs were isolated from cord blood using Ficoll-isopaque density gradient centrifugation [28,41,67,68] and were washed in Roswell Park Memorial Institute (RPMI) 1640 cell culture media (Gibco, Grand Island, NY, USA) twice to remove platelets and Ficoll. Cord blood does contain immature cells, including nucleated red blood cells, which a recent study revealed may also have an immune response [69]. As the purpose of this pilot study was to observe the immune response within the newborn period, we did not remove nucleated red blood cells from the suspension. The cells were then suspended in a freshly made freezing medium containing 90% Fetal Bovine Serum and 10% Dimethyl Sulfoxide (DMSO, Invitrogen, Carlsbad, CA, USA). After 24 h in a slow-freeze container at −80 °C, they were stored in liquid nitrogen until use.

Two independent experiments were run, each with 3–5 control and 3–5 PAE samples. For each experiment, samples were thawed in a water bath at 37 °C and washed twice in cRPMI (88% RPMI-1640, Gibco, Grand Island, NY, USA; 10% FBS, Gibco, Grand Island, NY, USA; 1% Penicillin/Streptomycin, Gibco, Grand Island, NY, USA; 1% Glutamax, Gibco, Grand Island, NY, USA) to remove the freezing medium. Live PBMCs were counted on a hemocytometer with trypan blue exclusion criteria, plated at a density of 0.2 million cells per well in batch one (1st baseline measurements) and 0.14 million cells per well in batch two (2nd baseline measurements) in a 24-well tissue culture plate and incubated at 37 °C, 5% CO_2_, and 8% Oxygen. Different cell densities were utilized to optimize the concentration of live cells present. After resting the cells overnight, the plates were centrifuged (300× *g* and 37 °C) and the media was replaced with cRPMI containing various TLR agonists including LPS (TLR4 agonist; 1 μg/mL; Sigma-Aldrich, Saint Louis, MO, USA), PAM3CSK4 (TLR2 agonist; 1 μg/mL; InvivoGen, San Diego, CA, USA), CpG ODN (TLR9 agonist; 1 μg/mL; InvivoGen, San Diego, CA, USA), or Poly IC (TLR3 agonist; 10 μg/mL; InvivoGen, San Diego, CA, USA) antigens separately, or cRPMI media (control). Just before adding the TLR agonists, a sample of the supernatant was collected and used as the baseline measurement. After 3 and 24 h post-stimulation with the same agonist [28], the supernatant from each well was collected, and stored at −80 °C. Concentrations of TLR agonists and the time points were chosen based on prior studies on human PBMCs [28].

### 4.3. Multiplex Immunoassay to Measure Functional Responses of PBMCs

Cytokine and chemokine levels were analyzed from the media supernatant from unstimulated control and stimulated PBMCs (1:4 dilution) using a V-plex Pro-inflammatory Panel 1 Human Kit (MesoScale Discovery, Gaithersburg, MD, USA). Protein levels of the following cytokines were measured: interferon-gamma (IFN-γ), interleukin-10 (IL-10), interleukin-12p70 (IL-12p70), interleukin-13 (IL-13), interleukin-1beta (IL-1β), interleukin-2 (IL-2), interleukin-4 (IL-4), interleukin-6 (IL-6), interleukin-8 (IL-8), and tumor necrosis factor alpha (TNF-α). The following cytokines are pro-inflammatory: IL-12p70, IL-6, IFN-γ, IL-2, IL-1β, TNF-α, and IL-8, with IL-13, IL-10, and IL-4 being anti-inflammatory. The cytokine levels were measured at baseline, prior to any stimulation with a TLR agonist. All these cytokines were also measured at time 24 h following stimulation with one of the four TLR agonists. Cytokine levels were also measured at the 3-h timepoint in the cells stimulated with the TLR4 agonist. Samples from each experiment were run simultaneously with appropriate standards. The plates were read on a QuickPlex SQ 120 Imager and the cytokine levels were reported as pg/mL (see Figure 1, Figure 2, Figure 3 and Figure 4). While all cytokines were measured, the figures included highlight the most interesting results of the cytokine levels following each TLR stimulation. Cytokine levels are described as percent change, which is the change in the level at 24 h compared to the baseline level for each cytokine and TLR simulation.

### 4.4. Statistical Analysis

Descriptive statistics were used to summarize continuous (*t*-test and Mann–Whitney test) and categorical (Chi-square and Fisher’s exact) maternal and infant characteristics and compare them among PAE and control study groups. Baseline measurements of cytokine levels in the control and PAE study groups were completed using Student’s *t*-test. The cytokine level was then assessed as a percent change from the baseline, which was calculated by taking the stimulated level measurement minus the baseline level, then dividing by the stimulated measurement and multiplying by 100. The percent change of cytokine levels was compared between the study groups using Student’s *t*-test for each individual cytokine; the representative figures for percent change are provided in Figure 1. The time comparison of cytokine levels following TLR4 stimulation was analyzed using a two-way analysis of variance (ANOVA) to assess differences between the groups (control vs. PAE) and across the time course (the baseline, 3 h, 24 h) following TLR4 stimulation. Post-hoc pairwise comparisons were conducted with the Tukey adjustment. In all analyses, statistical significance was defined at an alpha level of *p* < 0.05.

## 5. Conclusions

This study has revealed a unique cytokine level profile in umbilical cord blood following light-to-moderate PAE, with vulnerabilities noted in specific pathways, especially in TLR2 and TLR4 stimulation. Noting alterations in pro-inflammatory and anti-inflammatory cytokine levels in cord blood samples may indicate neuroinflammatory changes. This study lays the foundation for further investigations to characterize alterations in these pathways. The cytokine profiles and imbalance could be studied in relation to infant neurodevelopmental and immunological outcomes to assess the utility of cytokine profiles as predictive factors for more distant outcomes.

## Figures and Tables

**Figure 1 ijms-25-07019-f001:**
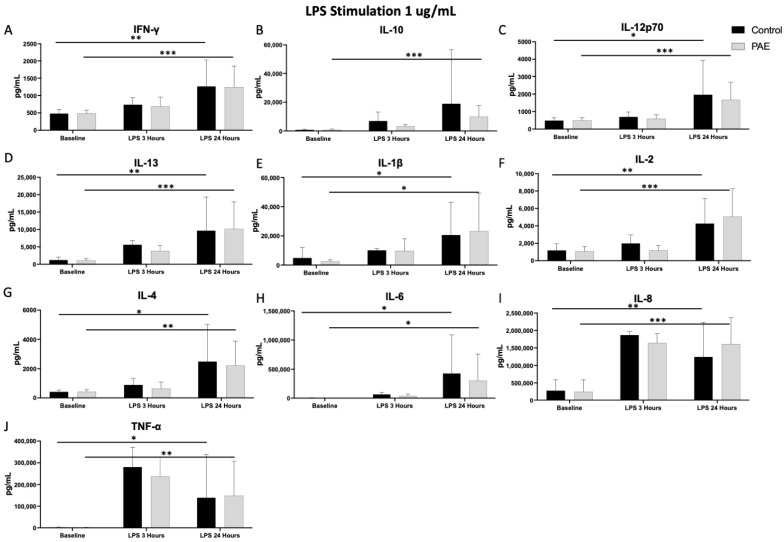
The Cytokine Expression Following TLR4 Agonist Stimulation. Figure 1 shows the mean cytokine expression for the control and PAE groups after stimulation with the TLR4 agonist, lipopolysaccharide (LPS). There was a significant difference in the increased expression of IL-2 (*p* < 0.05), IL-4 (*p* < 0.05), and IL-6 (*p* < 0.05) between the control and PAE groups from baseline to 3 h and 24 h. (**A**) Additionally, significant increases in IFN-γ (45% increase in controls and 49% increase in PAE; *p* < 0.001), (**C**) IL-12p70 (49% increase in controls and 57% increase in PAE; control: *p* < 0.05, PAE: *p* < 0.001), (**D**) IL-13 (59% increase in controls and 74% increase in PAE; control: *p* < 0.01, PAE: *p* < 0.001), (**E**) IL-1β (48% increase in controls and 71% increase in PAE; *p* < 0.05), (**F**) IL-2 (52% increase in controls and 66% increase in PAE; control: *p* < 0.01, PAE: *p* < 0.001), (**G**) IL-4 (58% increase in controls and 66% increase in PAE; control: *p* < 0.05, PAE: *p* < 0.01), (**H**) IL-6 (78% increase in controls and 86% increase in PAE; *p* < 0.05), (**I**) IL-8 (67% increase in controls and 72% increase in PAE; control: *p* < 0.01, PAE: *p* < 0.001) and (**J**) TNF-α (69% increase in controls and 83% increase in PAE; control: *p* < 0.05, PAE: *p* < 0.01) expression were noted in both control and PAE groups 24 h after stimulation with LPS. (**B**) IL-10 expression was only significantly increased in the PAE group from baseline to 24 h after stimulation (*p* < 0.001). (* *p* < 0.05, ** *p* < 0.01, *** *p* < 0.001; *n* = 8–10 per group, mean value ± SEM).

**Figure 2 ijms-25-07019-f002:**
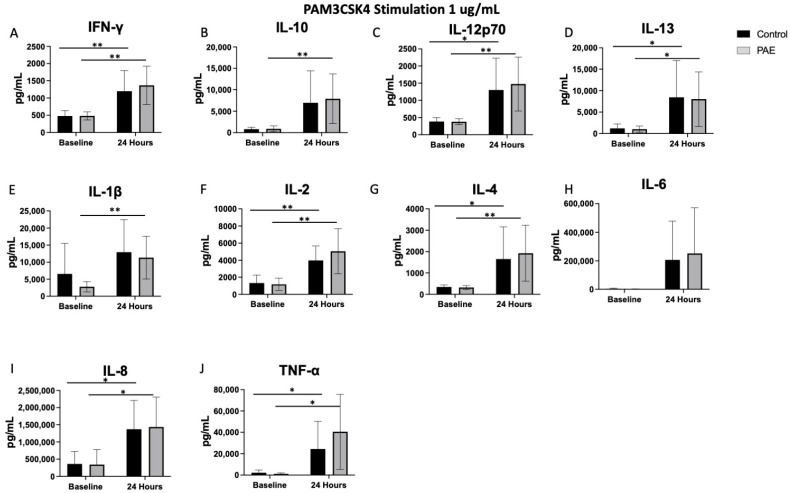
The Cytokine Expression Following TLR2 Agonist Stimulation. Figure 2 shows the mean cytokine expression for the control and PAE groups after stimulation with the TLR2 agonist, PAM3CSK4. There were significant increases in (**A**) IFN-γ (45% increase in controls and 56% increase in PAE; *p* < 0.01), (**C**) IL-12p70 (46% increase in controls and 62% increase in PAE; control *p* < 0.05, PAE *p* < 0.01), (**D**) IL-13 (62% increase in controls and 75% increase in PAE; *p* < 0.05), (**F**) IL-2 (57% increase in controls and 67% increase in PAE; *p* < 0.01), (**G**) IL-4 (46% increase in controls and 71% increase in PAE; control *p* < 0.05, PAE *p* < 0.01), (**I**) IL-8 (78% increase in controls and 72% increase in PAE; *p* < 0.05), and (**J**) TNF-α (69% increase in controls and 84% increase in PAE; *p* < 0.05) expression noted in both control and PAE groups 24 h after stimulation with PAM3CSK4. (**E**) IL-1β and (**B**) IL-10 had a significant increase only in the PAE group (*p* < 0.01). (**H**) IL-6 did not have any significant change (* *p* < 0.05, ** *p* < 0.01, *n* = 8–10 per group, mean value ± SEM).

**Figure 3 ijms-25-07019-f003:**
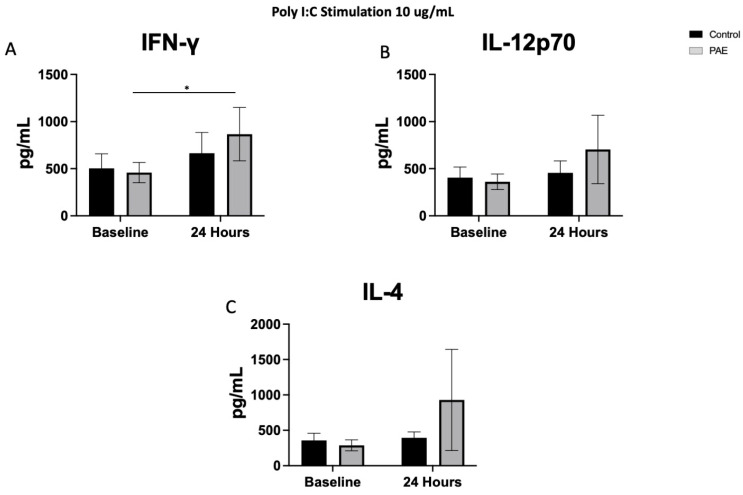
The Cytokine Expression Following TLR3 Agonist Stimulation. Figure 3 shows the mean cytokine expression for the control and PAE groups after stimulation with the TLR3 agonist, Poly I:C. Interestingly, there were increases in the expression of (**A**) IFN-γ only in the PAE group (23% increase in controls and 43% increase in PAE; *p* < 0.05). (**B**) IL-12p70 (11% increase in controls and 39% increase in PAE; *p* < 0.05) and (**C**) IL-4 (10% increase in controls and 53% increase in PAE; *p* < 0.05) had significantly different percent change of expression in the control group compared to PAE 24 h after stimulation with Poly I:C. (* *p* < 0.05, *n* = 8–10 per group, mean value ± SEM).

**Figure 4 ijms-25-07019-f004:**
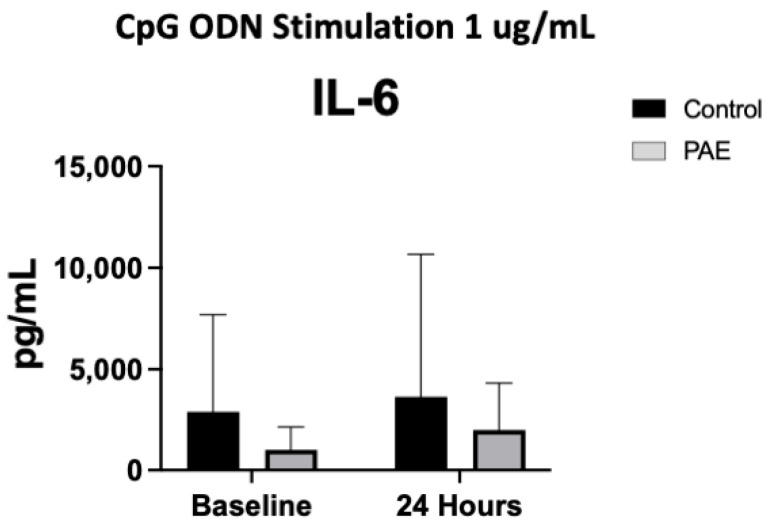
The Cytokine Expression Following TLR9 Agonist Stimulation. Figure 4 shows the mean cytokine expression for the control and PAE groups after stimulation with the TLR9 agonist, CpG ODN. Interestingly, there was a significant difference in the percent change of the control groups compared to the PAE group in the expression of IL-6 (5% increase in controls compared to 40% increase in PAE, *p* < 0.05) 24 h after stimulation with CpG ODN. (*n* = 8–10 per group, mean value ± SEM).

**Figure 5 ijms-25-07019-f005:**
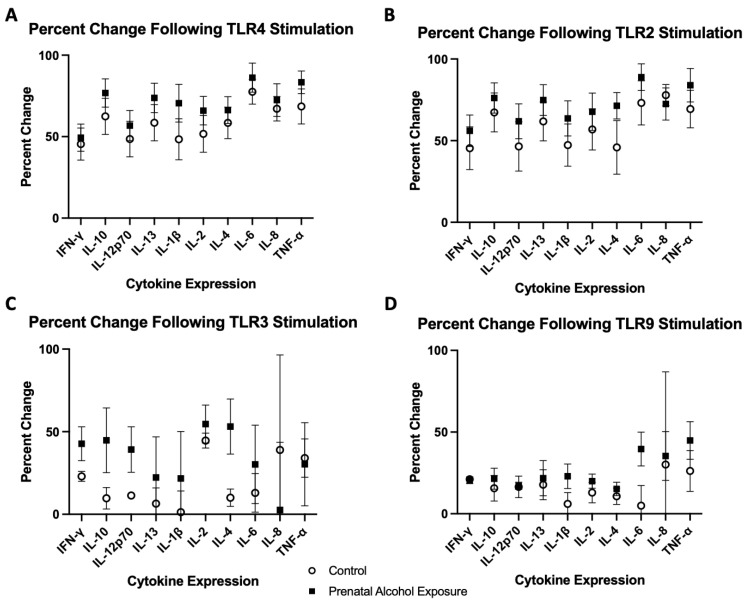
The percent change of cytokine levels following stimulation with various TLR agonists was calculated as the change at 24 h compared to the baseline measurement. (**A**) TLR4 and (**B**) TLR2 agonist stimulation resulted in higher percent changes of cytokine expression overall compared to the precent change in cytokine expression following (**C**) TLR3 and (**D**) TLR9 stimulation.

**Table 1 ijms-25-07019-t001:** Sociodemographic characteristics of the study participants stratified by study group (N = 18).

Variable	Healthy Control (N = 10)	PAE (N = 8)	*p*-Value
	Mean ± SD	Mean ± SD	
Maternal age	27.0 ± 6.1	27.5 ± 4.9	0.85 ^0^
Years of education	13.3 ± 1.3	14.3 ± 2.7	0.65 ^1^
Birthweight (grams)	3028.5 ± 547.8	3382.3 ± 286.8	0.12 ^0^
	N (%)	N (%)	
Marital status:			0.043 ^2^
Single/separated/divorced	1 (10.0%)	5 (62.5%)	
Married/cohabitating	9 (90.0%)	3 (37.5%)	
Race:			0.55 ^2^
White	5 (50.0%)	4 (50.0%)	
Black or African American	1 (10.0%)	0 (0.0%)	
American Indian	3 (30.0%)	1 (12.5%)	
Other	1 (10.0%)	3 (37.5%)	
Education level:			0.57 ^2^
High school or less	5 (50.0%)	3 (37.5%)	
Some college or vocational school	4 (40.0%)	2 (25.0%)	
College degree or higher	1 (10.0%)	3 (37.5%)	
Preterm delivery (<37 weeks)	1 (10.0%)	0 (0.0%)	1.00 ^2^
Infant sex: female	6 (60.0%)	2 (25.0%)	0.19 ^2^
Type of delivery: vaginal	8 (80.0%)	8 (100.0%)	1.00 ^2^
	Mean ± SD	Mean ± SD	
AAD ^a^	0.00 ± 0.00	0.45 ± 0.19	<0.0001 ^0^
AADD ^a^	0.00 ± 0.00	1.82 ± 0.44	<0.0001 ^0^

^a^ AAD, absolute alcohol (ounces) per day [1 AA is equivalent to ~2 standard drinks, so 1 AAD is equivalent to ~2 standard drinks per day]; AADD, absolute alcohol per drinking day; ^0^ based on pooled variances *t*-test; ^1^ based on Mann–Whitney test; ^2^ based on Fisher’s exact test.

## Data Availability

Data available on request due to privacy/ethical restrictions.

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
