# Peer review of "Moderate Prenatal Alcohol Exposure Increases Toll-like Receptor Activity in Umbilical Cord Blood at Birth: A Pilot Study"

_ijms, 2024, doi:10.3390/ijms25137019_

Round 1

Reviewer 1 Report (Previous Reviewer 2)

Comments and Suggestions for Authors

Dear authors,

Thanks for the paper that is submitted, which covers an interesting topic. I think this article attempts to explain the relationship between prenatal alcohol exposure and Toll-like receptor activity in the umbilical cord at birth. The article is well written and flows easily. I have major comments and edits that I would like to suggest to the authors:

In the Methods section, I suggest introducing the variables in one of the sections. For example, I think years of education is less clear than the educational level.

Please, check the journal s recommendations, in the citations, Tables and Figures. All references do not follow the ACS style guide.

Author Response

Reviewer 2 Report (New Reviewer)

Comments and Suggestions for Authors

After reviewing the manuscript's contents - below I present a brief review commentary of the submitted paper.

All major sections, the Introduction, Methodology, Results discussed and the Discussion - are presented in a clear and comprehensible manner for the reviewer.

Below are only a few minor comments.

Peripheral Blood Mononuclear Cell (PBMC) Isolation and Stimulation in vitro

Please provide manufacturer name and catalog number of reagents: FBS, RPMI-1640, cRPMI, DMSO, Penicillin/Streptomycin, Glutamax, as well as mentioned ligands: LPS, PAM3CSK4, CpG ODN, Poly IC. 

Why was only LPS used in the time-dependence evaluation scheme (that is, after 3 and 24 h, respectively), while the effects of the others were evaluated only after 24 h?.

Multiplex Immunoassay to Measure Functional Responses of PBMCs

Please write whether the same type of agonist was used in both cases (3h, 24h), please give its name (although this is evident from the information presented in the figures - but its name should appear in the text).

At the end of the discussion, maybe the authors briefly highlighted more the limitations of the evidence presented, and consequently the conclusions, maybe more in a holistic way? This is just so for the authors to consider.     

Author Response

Reviewer 3 Report (New Reviewer)

Comments and Suggestions for Authors

Esteemed authors and editorial team, this is a very interesting article regarding fetal inflammatory injury in the context of prenatal alcohol exposure. 

The study design is very well conducted, as well as the statistical analysis.

References are fit and up-to-date. Conclusions sound.

I merely suggest to emphasize there were no other potential confounding factors, other than the drug exposure already listed, which could lead to fetal proo-inflammatory reactions.

Author Response

Reviewer 4 Report (New Reviewer)

Comments and Suggestions for Authors

The article submitted by Maxwell et al. measured and compared cytokine and/or chemokine levels in the media supernatant from cultured peripheral blood mononuclear cells (PBMCs) isolated from umbilical bord bloods in response to agonists of TLRs in prenatally alcohol exposed (PAE) and control mothers at delivery. I feel the authors would like to identify cytokines/chemokines released from cultured PBMC in response to agonists to different types of TLRs in the PAE group. However, there have severe problems in this paper in terms of inappropriate use of statistical procedures and the interpretations of statistical analysis. These undermine the reliability of the findings.

Major concerns

1) Although cytokine/chemokine levels in the media are expressed as “pg/mL” in Figures, the levels are expressed as “% of baseline” in description of the Results. This inconsistency will lead the readers’ confusions. It is unclear details of procedures for sampling the media for the baseline measurements. So, I cannot determine whether % of baseline” is appropriate parameters. Please provide.

2) Cytokine/chemokine levels in Figures 1-4 are compared between the baselines and 3 and/or 24 hours after exposure to TLR agonists. Statistical differences of cytokine/chemokine levels should be compared between PAE and control groups in each time points. Statistical procedures used are different depending on whether cytokine/chemokine levels are expressed as pg/mL” of releasing into the cultured media or % of the baseline. At least, two-way ANOVA should be used in the TLR4 agonist stimulation experiment.

3) Tukey test is used as post-hoc pairwise comparisons in the TLR4 agonist stimulation experiment. As seen in bar graphs with error bars shown in Figure 1, the population variances for each group may be heterogeneous in some data. Please check it again.

4) Figure legends are too long and too complex. The legends should include the title of each subfigure, a summary of statistical results and abbreviations.

Minor points:

1) The authors should explain clearly the reasons why mean ± SD was used in Table 1 and why mean ± SEM was used in Figures 1-4.

2) The position of “Materials and Methods” in the text are not match for the IJMS style. Please see the instruction for authors and correct it.

Comments on the Quality of English Language

Please improve your technical expressions.  For example, "expression" usually refers to the production of a protein (translation) or RNA (transcription) from a gene. Cytokines/chemokines released into the media include not only those produced by translation but also those stored in PBMCs. Cytokine/chemokine “level(s)” would be appropriate term to use.

Round 2

Reviewer 4 Report (New Reviewer)

Comments and Suggestions for Authors

The manuscript was not revised to reflect my comments regarding the statistical analysis, particularly in interpretations of two-way ANOVA results and use of Tukey test as poet-hoc testing. The authors should mention TLR agonist-related time-dependent changes in cytokine/chemokine levels based on two-way ANOVA results, but did not.

Author Response

We sincerely apologize, as the prior summary we included that the ANOVA did not have significant results between 0, 3 and 24 hours was inadvertently deleted in the revision. The only significance noted was between time 0 and 24 hours, as previously included. We have now included this summary sentence. 

This manuscript is a resubmission of an earlier submission. The following is a list of the peer review reports and author responses from that submission.

Round 1

Reviewer 1 Report

Comments and Suggestions for Authors

The Paper aims to test whether the low-to-moderate level of PAE can affect fetal innate immune activity in humans. The main contribution of this manuscript is the utilization of multiple TLR agonists to characterize PAE-induced alterations by examining a more comprehensive profile of pro- and anti-inflammatory cytokines. The strength of the manuscript that the author claimed is that it tests the effect of low-to-moderate levels of PAE on fetal innate immune activity in humans while many studies were focused on moderate-to-high levels of PAE. However, the lack of scientific rigor has lowered the enthusiasm for the manuscript. Specifical, it reflects in the following aspects:

1. Study Population

For Exclusion Criteria: the exclusion of immunity-related sociodemographic and medical characteristics like history of autoimmune diseases and the use of immunosuppressive medications were not reported.

2. Data Collection

Alcohol Exposure Assessment: the categorization of PAE into Low-to-moderate exposure, and the verification of self-reports with biomarkers were not reported in the results (e.g., gamma-glutamyl transpeptidase). It is the key issue that if not addressed, might lead to irreproducibility.

3. Data analysis

a) The analyses restricted to confirmed cases of alcohol consumption through biomarkers were not reported. The reliance on self-reported alcohol consumption may introduce bias.

b) The observational experimental results cannot establish evidence and lead to a conclusion.

c) lack of rigor in experimental design, for example, the choice of the time course tested and the choice of the dose of the multiple TLR agonists. Inconsistency in the tested time course for some of the other TLR agonists compared with the TLR4 agonist.

4. Citation

The relevance of Cited References is not accurate. For example, in text line 107, citation 21. 

Author Response

We greatly appreciate the opportunity to edit the manuscript and strengthen the article based on your recommendations.

  1. Study Population

For Exclusion Criteria: the exclusion of immunity-related sociodemographic and medical characteristics like history of autoimmune diseases and the use of immunosuppressive medications were not reported.

While not an exclusion criterion, information on autoimmune disorders were collected via self-report. No individuals in this pilot study reported having any autoimmune disorders.

  1. Data Collection

Alcohol Exposure Assessment: the categorization of PAE into Low-to-moderate exposure, and the verification of self-reports with biomarkers were not reported in the results (e.g., gamma-glutamyl transpeptidase). It is the key issue that if not addressed, might lead to irreproducibility. 

We apologize for this oversite. We have added the average alcohol per day intake and the average alcohol per drinking day to Table 1, which was obtained through prospective repeated self-report, including TLFB measures and targeted questions about binge drinking.

  1. Data analysis
  2. a) The analyses restricted to confirmed cases of alcohol consumption through biomarkers were not reported. The reliance on self-reported alcohol consumption may introduce bias.

We appreciate this comment and have added this point to the limitations of the study.

  1. b) The observational experimental results cannot establish evidence and lead to a conclusion. 

We appreciate this comment and emphasize now that causality cannot be establish from this preliminary report.

  1. c) lack of rigor in experimental design, for example, the choice of the time course tested and the choice of the dose of the multiple TLR agonists. Inconsistency in the tested time course for some of the other TLR agonists compared with the TLR4 agonist. 

We appreciate this comment. We purposefully chose the time course for the TLR4 agonist given the depth of knowledge on timing of response following stimulation with TLR4 agonist. The remainder of TLR agonists had collection of samples at 24 hours to explore initial responses in this pilot study. We have added the future direction of expanding the time course and the dose.

  1. Citation

The relevance of Cited References is not accurate. For example, in text line 107, citation 21.

We apologize for this error and have updated this citation with the correct reference.

Reviewer 2 Report

Comments and Suggestions for Authors

Dear authors,

Thanks for the paper that is submitted, which covers an interesting topic. I think this article attempts to explain the relationship between prenatal alcohol exposure and Toll-like receptor activity in the umbilical cord at birth. The article is well written and flows easily. I have major comments and edits that I would like to suggest to the authors:

Abstract: The extension is too long and does not follow the requirements of the journal. I suggest shortening the abstract and remove lines 12-16.

Methods:

Although the original study research protocol was approved by the University of Mexico (I assume is approvedby an ethics committee), I have some concerns about knowing the PEA in pregnant women without implementing preventive strategies. There is no reference in the article on this issue, so I would like to know if these preventive strategies were taken into account.

Results:

Lines 210-212: The way in which alcohol consumption is calculated is unclear. It should be necessary to introduce more information in the Method Section.

I think a table with the alcohol consumption should be included.

Discussion:

Line 321: Control group.

Please, check the journal s recommendations, in the citations, Tables and Figures. All references do not follow the ACS style guide.

The authors Maxwell and Noor have 3 self-citations. I suggest economising them and using them if necessary.

Author Response

We appreciate the reviewer comments as these edits have improved the manuscript. We hope the reviewer finds these changes to be satisfactory.

Abstract: The extension is too long and does not follow the requirements of the journal. I suggest shortening the abstract and remove lines 12-16.

We have reviewed the journal requirements and see that the above should “be a total of about 200 words maximum”. We have shortened the abstract to meet this requirement.

Methods:

Although the original study research protocol was approved by the University of Mexico (I assume is approved by an ethics committee), I have some concerns about knowing the PEA in pregnant women without implementing preventive strategies. There is no reference in the article on this issue, so I would like to know if these preventive strategies were taken into account.

This study was approved by the Institutional Review Board (included in methods description). All participants (regardless of the disclosed amount of alcohol use) received counseling about risks associated with alcohol use in pregnancy and were provided with resources, including a counseling hotline. This information has been added to the Methods.

Results:

Lines 210-212: The way in which alcohol consumption is calculated is unclear. It should be necessary to introduce more information in the Method Section.

We have included this information in the Materials and Methods section to be transparent in the methodology of the alcohol consumption calculation.

I think a table with the alcohol consumption should be included.

As requested, this information has been added to Table 1.

Discussion:

Line 321: Control group.

This typo has been corrected.

Please, check the journal s recommendations, in the citations, Tables and Figures. All references do not follow the ACS style guide.

The references have been re-formatted using EndNote to the IJMS format. Tables and figures have been reviewed to ensure compliance with the journal formatting.

The authors Maxwell and Noor have 3 self-citations. I suggest economising them and using them if necessary.

We have included self-citations only to support methodologies and have removed self-citations from the introduction and discussion sections.

Round 2

Reviewer 1 Report

Comments and Suggestions for Authors

Abstract

1. The author aims to test the effect of low-to-moderate levels of PAE on fetal immune activity in humans while many studies were focused on moderate-to-high levels of PAE. Authors hypothesized that (low-to-moderate?) PAE would impair immune responses. To test the central hypothesis, the authors employed umbilical cord-derived blood PMBCs to sought to determine if alterations in cytokine expression could be measured at the time of delivery in infants with (low-to-moderate?) PAE and controls. The authors hypothesized that “PAE would result in TLR-mediated dysregulated cytokine production in umbilical cord blood-derived leukocytes. While there is prior evidence from human fetal lymphocytes in vivo and in vitro studies and a growing body of preclinical data focusing on alcohol and potential TLR4 interactions, the effects of PAE on functional responses on other TLRs are relatively unknown. Thus, multiple TLR agonists were utilized to better characterize PAE-induced alterations through examining a more comprehensive profile of pro- and anti-inflammatory cytokines”. The abstract should clearly articulate these hypotheses and the research gap.

2. The abstract does not discuss the results, a critical omission that needs addressing.

3. The authors claimed that “Results of this investigation and additional studies could further inform novel biological mechanisms underpinning adverse outcomes in infants with PAE as well as potentially lead to identification of novel prognostic biomarkers”. The term “novel biological mechanisms” is mentioned but not explained. Clarification on what these mechanisms entail would enhance understanding.

Rigor in experimental design

The experimental design lacks scientific rigor, notably in the choice to examine cytokine profiles following stimulation by TLR3 and TLR9 agonists and the inconsistency in cytokine tests across different TLR stimulations. Additionally, measuring the expression levels of the tested TLRs is crucial, as these levels could influence the stimulatory response.

Material and Methods

1: while the low-to-moderate PAE reference range is not reported, hindering the ability to categorize alcohol exposure grouping. The reference range will help to understand how did authors group different alcohol exposure and to give a clear comparison of low-to-moderate alcohol exposure with high alcohol exposure. The absence of a reported reference range for low-to-moderate PAE is a significant gap,

2: There seems to be a discrepancy in the Timeline Follow-Back (TLFB) interviews' reported periods. Detailed TLFB time points and the procedure for assessing alcohol exposure should be provided for clarity. For instance, the authors claimed that “Alcohol use in the periconceptional period and during pregnancy was ascertained via self-report in four Timeline Follow-Back (TLFB) interviews [27]”, and cited reference 27. In reference 27, the period of TLFB approach as described in the sentence “the four study visits (i.e., prenatal, at delivery/birth, at 6 months postpartum, and at 20 months postpartum)” is not consistent with what author reported here in the manuscript. If they are not the same TLFB approach, please provide the detailed TLFB timepoints and procedure to assess and alcohol exposure assessment period used in this manuscript.

3: Information on how participants were divided into groups based on their standard drink units (SDU) is vague. Detailed grouping criteria and SDU for each group are necessary. Authors claimed that participants who demonstrated “<13 standard drink units (SDU) in the LMP” were grouped into be “control group” but did not show the number of SDU in the PAE group. Please give detailed information to show how the authors grouped the Participants into two groups and to show the number of SDU for each group.  

4: The manuscript claims ethanol biomarkers were used to categorize PAE but does not report these findings in the results. This discrepancy requires correction, especially if the study relies solely on self-reporting without biomarker confirmation. For instance, the authors claimed that “categorize PAE into low-to-moderate PAE group based on interview findings and ethanol biomarkers from biological specimens (page 3)” and “comprehensive ethanol biomarker panels were evaluated in the mother at enrollment and delivery (gamma-glutamyl transpeptidase [GGT], carbohydrate-deficient transferrin [CDT], phosphatidyl ethanol [PEth], urine ethyl glucuronide and ethyl sulfate [uEtG/uEtS]) and in the infant at delivery (PEth in dry blood spots) [28]”. The verification of self-reports with biomarkers was neither reported in the results (e.g., gamma-glutamyl transpeptidase) nor in the cited report in reference 28. If this study were based on self-report only, without confirmation from the biomarkers of the parent study, then this section needs to be corrected.

Statistical analysis

The methodology for assessing cytokine expression as a percent change from baseline does not align with the presentation of results, leading to confusion. If percent changes were analyzed, this data should be included as supplemental materials. i.e., the reported “The cytokine expression was then assessed as a percent change from the baseline, which was calculated by taking the stimulated expression measurement minus the baseline expression, then dividing by the stimulated measurement and multiplying by 100. The percent change of cytokine expression was compared between the study groups using a Student’s t-test for each cytokine” method is not consistent with the data figures shown in the “Results”. I would assume that the Y axis of the % change would be used if it is true. For example, figure 1 shows the mean cytokine expression for the control and PAE groups in the Y axis rather than % change. If the % changes were calculated and compared in other figures, please supply the data as supplemental materials.  

Results

1. Authors primarily reported outcomes of differences in levels of immune markers, differences in cell profiles like cell viability and purity in cord blood among the groups were not mentioned as they are critical to response to the agonists stimulation.

2. The interpretation of results is insufficient, failing to convey the significance of the findings. Please address this issue.

3. Units for alcohol exposure are missing in Table 1, and a consistent method for assessing alcohol exposure is needed. Please provide more information to address how authors calculated the level of alcohol exposure and defined it as light-to-moderate in the PAE group. Please use the consistent units to assess alcohol exposure when grouping the participants as well.

4. In all figures, if the % changes were calculated and compared, please supply the data figures that show % changes in y-axis as supplemental materials. 

Conclusion

The study's conclusion mentions vulnerabilities in specific pathways following light-to-moderate PAE without specifying these pathways. Providing details would enrich the conclusion.

Author Response

Abstract

  1. The author aims to test the effect of low-to-moderate levels of PAE on fetal immune activity in humans while many studies were focused on moderate-to-high levels of PAE. Authors hypothesized that (low-to-moderate?) PAE would impair immune responses. To test the central hypothesis, the authors employed umbilical cord-derived blood PMBCs to sought to determine if alterations in cytokine expression could be measured at the time of delivery in infants with (low-to-moderate?) PAE and controls. The authors hypothesized that “PAE would result in TLR-mediated dysregulated cytokine production in umbilical cord blood-derived leukocytes. While there is prior evidence from human fetal lymphocytes in vivo and in vitro studies and a growing body of preclinical data focusing on alcohol and potential TLR4 interactions, the effects of PAE on functional responses on other TLRs are relatively unknown. Thus, multiple TLR agonists were utilized to better characterize PAE-induced alterations through examining a more comprehensive profile of pro- and anti-inflammatory cytokines”. The abstract should clearly articulate these hypotheses and the research gap.

We appreciate this feedback and have added as much as possible to the 200-word count abstract to more clearly articulate the hypothesis and research gaps as suggested above.

  1. The abstract does not discuss the results, a critical omission that needs addressing.

Due to the word limitation, not all the detailed results can be included in the abstract. We included at least 1 finding per TLR agonist stimulation experiment to provide the breadth of results.

  1. The authors claimed that “Results of this investigation and additional studies could further inform novel biological mechanisms underpinning adverse outcomes in infants with PAE as well as potentially lead to identification of novel prognostic biomarkers”. The term “novel biological mechanisms” is mentioned but not explained. Clarification on what these mechanisms entail would enhance understanding.

We apologize for the confusion created with the use of this terminology and have deleted this phrase for clarification.

Rigor in experimental design

  1. The experimental design lacks scientific rigor, notably in the choice to examine cytokine profiles following stimulation by TLR3 and TLR9 agonists and the inconsistency in cytokine tests across different TLR stimulations. Additionally, measuring the expression levels of the tested TLRs is crucial, as these levels could influence the stimulatory response.

The reviewer expressed concerns about the inconsistency in cytokine tests across different TLR stimulations. All the cytokines listed were tested for each TLR stimulation. To keep the figures more simple and easier to interpret, only those with differences were included in the figures. We have added a sentence to the methods to clarify that all cytokines were measures for all TLR simulations. The expression levels of the tested TLRs were obtained, with the figures showing the results in pg/mL. To make the comparisons easier for interpretation, the percent change was used in the text of the manuscript, with these figures now added as supplemental material.

In regards to choosing TLR3 and TLR9 agonists, this was done given the limited literature available during the research design period. Given that TLR3 and TLR9 have immune responses, these agonists were included as the overarching hypothesis was that PAE would result in alteration to immune response, as observed by differences in cytokine expression following TLR agonist stimulation. We have added this clarification to the introduction to improve clarity as to the choices made.   

Material and Methods

1: while the low-to-moderate PAE reference range is not reported, hindering the ability to categorize alcohol exposure grouping. The reference range will help to understand how did authors group different alcohol exposure and to give a clear comparison of low-to-moderate alcohol exposure with high alcohol exposure. The absence of a reported reference range for low-to-moderate PAE is a significant gap,

The information on the alcohol exposure in this specific group is included in the results and is reported as equivalent to approximately 3 drinks per week. This information is just before Table 1. The overall commonly used definitions of low PAE (less than 3 drinks per week) and moderate PAE (less than 7 drinks per week) has been added to the methods and additional citations have been included to support this definition.

2: There seems to be a discrepancy in the Timeline Follow-Back (TLFB) interviews' reported periods. Detailed TLFB time points and the procedure for assessing alcohol exposure should be provided for clarity. For instance, the authors claimed that “Alcohol use in the periconceptional period and during pregnancy was ascertained via self-report in four Timeline Follow-Back (TLFB) interviews [27]”, and cited reference 27. In reference 27, the period of TLFB approach as described in the sentence “the four study visits (i.e., prenatal, at delivery/birth, at 6 months postpartum, and at 20 months postpartum)” is not consistent with what author reported here in the manuscript. If they are not the same TLFB approach, please provide the detailed TLFB timepoints and procedure to assess and alcohol exposure assessment period used in this manuscript.

The TLFB as referenced in 27 is the correct methodology. For this pilot study, the umbilical cord samples were obtained at delivery. Therefore, the 6 months postpartum and 20 months postpartum times had not occurred yet and were not included for this pilot study. The following sentence was added to the material and methods to clarify this point “As this study focuses on samples obtained at delivery, the remaining postpartum TLFB interviews obtained for the ENRICH-2 cohort study are not applicable for this pilot study.”

3: Information on how participants were divided into groups based on their standard drink units (SDU) is vague. Detailed grouping criteria and SDU for each group are necessary. Authors claimed that participants who demonstrated “<13 standard drink units (SDU) in the LMP” were grouped into be “control group” but did not show the number of SDU in the PAE group. Please give detailed information to show how the authors grouped the Participants into two groups and to show the number of SDU for each group.

The information on the SDU in each group is presented to Table 1. Both the average alcohol per day and the average alcohol per drinking day are included, with the control group having 0.00 SDU in both instances. The PAE group had a mean of 0.45 SDU per day and 1.82 SDU per drinking day. As expected, the amount of alcohol intake was significantly different between the two groups.

4: The manuscript claims ethanol biomarkers were used to categorize PAE but does not report these findings in the results. This discrepancy requires correction, especially if the study relies solely on self-reporting without biomarker confirmation. For instance, the authors claimed that “categorize PAE into low-to-moderate PAE group based on interview findings and ethanol biomarkers from biological specimens (page 3)” and “comprehensive ethanol biomarker panels were evaluated in the mother at enrollment and delivery (gamma-glutamyl transpeptidase [GGT], carbohydrate-deficient transferrin [CDT], phosphatidyl ethanol [PEth], urine ethyl glucuronide and ethyl sulfate [uEtG/uEtS]) and in the infant at delivery (PEth in dry blood spots) [28]”. The verification of self-reports with biomarkers was neither reported in the results (e.g., gamma-glutamyl transpeptidase) nor in the cited report in reference 28. If this study were based on self-report only, without confirmation from the biomarkers of the parent study, then this section needs to be corrected.

We appreciate the opportunity to clarify the group of study participants. The revised version now more clearly describes the initial grouping of participants into PAE and Control group based on self-reported alcohol use in periconceptional period and additional criteria for each group to maintain eligibility based on TLFB interviews later in pregnancy. In the main study, biomarkers were used as the third criterion to guide identification of subjects for neurodevelopmental assessment at 6 months of age. Since this manuscript only focuses on prenatal period, a subset of patients was chosen based on their self-reported alcohol use in periconceptional period and during pregnancy (prospective, repeated TLFB interviews, which are considered a ‘gold standard’ for assessment of PAE). As requested, information on ethanol biomarkers has been removed, and the TLFB interviewed are now described in greater details.

Statistical analysis

The methodology for assessing cytokine expression as a percent change from baseline does not align with the presentation of results, leading to confusion. If percent changes were analyzed, this data should be included as supplemental materials. i.e., the reported “The cytokine expression was then assessed as a percent change from the baseline, which was calculated by taking the stimulated expression measurement minus the baseline expression, then dividing by the stimulated measurement and multiplying by 100. The percent change of cytokine expression was compared between the study groups using a Student’s t-test for each cytokine” method is not consistent with the data figures shown in the “Results”. I would assume that the Y axis of the % change would be used if it is true. For example, figure 1 shows the mean cytokine expression for the control and PAE groups in the Y axis rather than % change. If the % changes were calculated and compared in other figures, please supply the data as supplemental materials.  

We apologize for this confusion and have created a supplemental figure to show the percent change data. We hope this additional figure will improve the reader’s overall understanding.  

Results

  1. Authors primarily reported outcomes of differences in levels of immune markers, differences in cell profiles like cell viability and purity in cord blood among the groups were not mentioned as they are critical to response to the agonists stimulation.

In the methods section we included information on live PBMC counts and the cell density used (lines 363-366). The live cells were counted on a hemocytometer to ensure consistency in samples.    

  1. The interpretation of results is insufficient, failing to convey the significance of the findings. Please address this issue.

We have added additional interpretations of the results to the results section to further convey the significance of the findings.

  1. Units for alcohol exposure are missing in Table 1, and a consistent method for assessing alcohol exposure is needed. Please provide more information to address how authors calculated the level of alcohol exposure and defined it as light-to-moderate in the PAE group. Please use the consistent units to assess alcohol exposure when grouping the participants as well.

Additional information on the alcohol levels has been added to Table 1, including the following “Calculated from TLFB interview in which participant was asked to recall alcohol consumption over the prior two weeks. Values presented are standard drink units. Two standard drinks equal one ounce of absolute alcohol.

  1. In all figures, if the % changes were calculated and compared, please supply the data figures that show % changes in y-axis as supplemental materials. 

We apologize for this oversight and have now added this as a supplemental figure. 

Conclusion

The study's conclusion mentions vulnerabilities in specific pathways following light-to-moderate PAE without specifying these pathways. Providing details would enrich the conclusion.

An additional sentence has been added to the conclusions to tie together the peripheral cytokine expression alterations with the possible neuroinflammatory changes, would could then impact neurodevelopmental and immunological outcomes.

Reviewer 2 Report

Comments and Suggestions for Authors

Dear authors,

Thank you for taking the time to address comments on the manuscript. The manuscript has been greatly improved, but there are still some concerns about some issues:

The way in which alcohol consumption is calculated is not yet clear. It is said in the article that the TLFB is used to assess alcohol consumption during pregnancy, but once the questionnaire is consulted, pregnant women must introduce the number of standard drinks. Did you have to train them to do so? It is not easy to do it yourself. 

In Table 1, the units for alcohol consumption should be introduced. 

Author Response

We are appreciative of the reviewer suggesting additional clarification is provided to the alcohol consumption.

We have added additional information to the methods, as well as to the Table with the drinking information to add additional clarity. We have also added an additional reference that utilized the TLFB technique. 

We hope that these changes clarify the alcohol consumption calculation. 

Round 3

Reviewer 1 Report

Comments and Suggestions for Authors

Title, Abstract, Introduction, and Rigor in experimental design

The reported findings or results cannot reflect the title. What is this pilot study’s crucial purpose? Is it acting as a preliminary investigation before embarking on a full-scale research project? What is its primary goal?

In the abstract, the interpretation of results is insufficient and unspecific, please convey the significance of the findings and be more specific. What are these preliminary data indicating in this pilot study? What are they serving for? Whether the research hypothesis is plausible and worth investigating further in a full-scale study.

The authors utilized multiple TLR agonists to better characterize PAE-induced alterations by examining a more comprehensive profile of pro- and anti-inflammatory cytokines. The Introduction did not discuss the rationale and scientific rigor of the examination of anti-inflammatory cytokines. What are pro-inflammatory cytokines? What are anti-inflammatory cytokines? Why did the authors choose these cytokines to test? The authors could not give a good explanation of why the anti-inflammatory cytokines and profiles were chosen to be examined. 

Measuring the expression levels of the tested TLRs is crucial, as these levels could influence the stimulatory response. In the author’s reply, the authors claimed that “The expression levels of the tested TLRs were obtained, with the figures showing the results in pg/mL”, while I could not find the figure. It indicated that additional experiments were needed.

The authors claimed that “To make the comparisons easier for interpretation, the percent change was used in the text of the manuscript, with these figures now added as supplemental material”. Given the percentage change used in the text of the manuscript, the authors failed to cite the supplemental figures in the text which will confuse readers.  

Material and Methods

In line 144-152, the authors compared and reported that the participants in the control group “were required to demonstrate no more than minimal self-reported alcohol use during the periconceptional period: <13 standard drink units 145 (SDU) and no binge drinking episodes (defined as 4 or more SDU on one occasion) in the month surrounding last menstrual period (LMP) and the participants in the PAE group based on reported perinatal alcohol consumption (of 2 binge drinking episodes or 13 SDU in the periconceptional period – the month surrounding LMP”. This information is vague. If this is important and detailed grouping criteria information, please indicate these data in Table 1 to give a direct comparison between the control group and the PAE group and use the same unit to keep consistency. In the author’s reply, the authors said “Both the average alcohol per day and the average alcohol per drinking day are included, with the control group having 0.00 SDU in both instances. The PAE group had a mean of 0.45 SDU per day and 1.82 SDU per drinking day. As expected, the amount of alcohol intake was significantly different between the two groups.” the unit is confusing in “0.00 SDU” and “0.45 SDU per day” as shown in AAD notes as ounces per day.

In line 236, “1AA is equivalent to 0.5 standard drunks” is confusing. What does AA represent? Is it per day or per week?

In line 174, more detailed information for the cord blood PBMC isolation. cited reference 23 is not an isolating protocol for cord blood but for peripheral blood. Cord blood contains immature cells, including nucleated red cells, which can result in significant contamination of the mononuclear cell and removal of these cells requires additional steps. There are differences between peripheral blood and cord blood due to the subpopulation distribution. The detailed protocol from cited reference 23 cannot be found, indicating that the reference authors cited is not relevant to the research. Please check the whole manuscript for reference relevance and accuracy. If it is the merged protocol from both the cited references 23 and 32, please provide the detailed revised protocol and materials associated with the isolation. 

Statistical analysis

The % changes were calculated and compared in supplemental figures but missing SD or SEM.

Results

In the author’s reply, it was claimed that “In the methods section we included information on live PBMC counts and the cell density used (lines 363-366). The live cells were counted on a hemocytometer to ensure consistency in samples.”, while I cannot find out this information in line 363-366.

The Results section failed to present the data in a clear, concise, and objective manner. For example, in lines 302-303, the authors claimed that “cytokine expression increased at 3 hours compared to the baseline…” which is not a strict and accurate report as there is no statistical increase at 3 hours compared to the baseline in IFN-γ action and most of the cytokines. The figure legend for the supplemental material was not well addressed. For example, at which time point (at 3-, or 24 hours?) the % changes were calculated? This is just one example; the problem exists in the Results section. 

The interpretation of results is still insufficient, failing to convey the significance of the findings.

Units for alcohol exposure (ounces of absolute alcohol per day) in Table 1 and units for definition as moderate in the PAE group (less than 7 drinks per week) are not consistent. A clearer report of the calculation and assessment is needed.

Author Response

We appreciate the opportunity to improve the manuscript based on the reviewer comments, and hope these changes adequately address any concerns. Please see below for the specific responses:

The reported findings or results cannot reflect the title. What is this pilot study’s crucial purpose? Is it acting as a preliminary investigation before embarking on a full-scale research project? What is its primary goal? 

The overall goal of this project was to determine if there were findings that would warrant a full-scale research project. Thus, the more notable impact on TLR2 and TLR4 suggest this would be the area to focus the next larger scale project. The abstract has been revised to reflect this as well as the body of the manuscript.

In the abstract, the interpretation of results is insufficient and unspecific, please convey the significance of the findings and be more specific. What are these preliminary data indicating in this pilot study? What are they serving for? Whether the research hypothesis is plausible and worth investigating further in a full-scale study.

The preliminary data in this pilot study support further work investigating TLR2 and TLR4 stimulation follow PAE. As these specific areas are researched, questions could be probed including if these alterations in the stimulation have clinical significance and if there are any associations with the TLR stimulation alterations and neurodevelopmental outcomes. Ultimately, if there is a “profile” of alterations identified, it could be used as a predictive tool. This is very far in the future, and we recognize this current pilot study cannot be used in that way, but rather support further investigations into TLR2 and TLR4. We have changed the abstract conclusion to better reflect this as well as throughout the manuscript.

The authors utilized multiple TLR agonists to better characterize PAE-induced alterations by examining a more comprehensive profile of pro- and anti-inflammatory cytokines. The Introduction did not discuss the rationale and scientific rigor of the examination of anti-inflammatory cytokines. What are pro-inflammatory cytokines? What are anti-inflammatory cytokines? Why did the authors choose these cytokines to test? The authors could not give a good explanation of why the anti-inflammatory cytokines and profiles were chosen to be examined.  

We apologize that we have not previously clearly defined the pro- and anti-inflammatory cytokines and the importance of these. We have added this to the introduction (and categorized the cytokines into pro- or anti-inflammatory in the methods) and hope this is now more clear to readers.

Measuring the expression levels of the tested TLRs is crucial, as these levels could influence the stimulatory response. In the author’s reply, the authors claimed that “The expression levels of the tested TLRs were obtained, with the figures showing the results in pg/mL”, while I could not find the figure. It indicated that additional experiments were needed. 

Figures 1, 2, 3, and 4 all show the expression of the TLRs in pg/mL. The “baseline” in the graphs shows the expression of the cytokine prior to the stimulation. We have presented only the more interesting or significant cytokines rather than showing graphs for all 10 cytokines for each TLR. We do have the graphs and can add if this is felt appropriate (we thought it may overwhelm readers and would not highlight the significant areas). If it felt that all graphs should be included, we can add those.

The authors claimed that “To make the comparisons easier for interpretation, the percent change was used in the text of the manuscript, with these figures now added as supplemental material”. Given the percentage change used in the text of the manuscript, the authors failed to cite the supplemental figures in the text which will confuse readers.  

We apologize for this oversight and have added the appropriate citation in the manuscript to appropriate refer the readers to the supplemental figures.

Material and Methods

In line 144-152, the authors compared and reported that the participants in the control group“were required to demonstrate no more than minimal self-reported alcohol use during the periconceptional period: <13 standard drink units 145 (SDU) and no binge drinking episodes (defined as 4 or more SDU on one occasion) in the month surrounding last menstrual period (LMP)” and the participants in the PAE group “based on reported perinatal alcohol consumption (≥of ≥2 binge drinking episodes or ≥13 SDU in the periconceptional period – the month surrounding LMP”. This information is vague. If this is important and detailed grouping criteria information, please indicate these data in Table 1 to give a direct comparison between the control group and the PAE group and use the same unit to keep consistency. In the author’s reply, the authors said “Both the average alcohol per day and the average alcohol per drinking day are included, with the control group having 0.00 SDU in both instances. The PAE group had a mean of 0.45 SDU per day and 1.82 SDU per drinking day. As expected, the amount of alcohol intake was significantly different between the two groups.” the unit is confusing in “0.00 SDU” and “0.45 SDU per day” as shown in AAD notes as ounces per day. 

We have provided additional clarification to SDU and AAD/AADD and added conversion values to aide in the clarity of the readers. These terms are commonly used in the prenatal alcohol field, and definitions are from published manuscripts, including those specific to the parent study, and information provided by the National Institute on Alcohol Abuse and Alcoholism (https://www.niaaa.nih.gov/health-professionals-communities/core-resource-on-alcohol/basics-defining-how-much-alcohol-too-much).

In line 236, “1AA is equivalent to 0.5 standard drunks” is confusing. What does AA represent? Is it per day or per week?

We appreciate the opportunity to clarify this point. 1 AA is fluid ounce of absolute alcohol, here fore, the term “AA” itself does not define a specific time. As noted in the table and Methods, the values we present are AAD (absolute alcohol per day) and AADD (absolute alcohol per drinking day).  

In line 174, more detailed information for the cord blood PBMC isolation. cited reference 23 is not an isolating protocol for cord blood but for peripheral blood. Cord blood contains immature cells, including nucleated red cells, which can result in significant contamination of the mononuclear cell and removal of these cells requires additional steps. There are differences between peripheral blood and cord blood due to the subpopulation distribution. The detailed protocol from cited reference 23 cannot be found, indicating that the reference authors cited is not relevant to the research. Please check the whole manuscript for reference relevance and accuracy. If it is the merged protocol from both the cited references 23 and 32, please provide the detailed revised protocol and materials associated with the isolation.  

We apologize for the confusion with the reference. The “reference 23” was used to ensure the appropriate media was used and to determine the concentrations of the TLR agonists for these experiments. Additional reference has been added with the Ficoll-isopaque protocol. While we appreciate that cord blood and adult blood have differing compositions, the goal of this project was to determine any differences in the response that would occur in the newborn. Given that nucleated red cells may be present for the first few days of life, we purposefully did not remove these cells from the isolation (as recent studies show they may also provide an immune response and therefore could contribute to the responses noted). We have added this clarifying point to the methods.

Statistical analysis

The % changes were calculated and compared in supplemental figures but missing SD or SEM.

The SEM has now been added to the supplemental figure. We appreciate the opportunity to correct this omission.

Results

In the author’s reply, it was claimed that “In the methods section we included information on live PBMC counts and the cell density used (lines 363-366). The live cells were counted on a hemocytometer to ensure consistency in samples.”, while I cannot find out this information in line 363-366. 

The line numbers provided by the reviewer are not matching with the current version, therefore we will provide a description of the location of this statement rather than a line number. In the methods section under the Peripheral Blood Mononuclear Cell (PMBC) Isolation and Stimulation in vitro, the information of live cells being counted on a hemocytometer is in the second paragraph in the second sentence – “Live PBMCs were counted on a hemocytometer with trypan blue exclusion criteria….”.

The Results section failed to present the data in a clear, concise, and objective manner. For example, in lines 302-303, the authors claimed that “cytokine expression increased at 3 hours compared to the baseline…” which is not a strict and accurate report as there is no statistical increase at 3 hours compared to the baseline in IFN-γ action and most of the cytokines. The figure legend for the supplemental material was not well addressed. For example, at which time point (at 3-, or 24 hours?) the % changes were calculated? This is just one example; the problem exists in the Results section.  

We agree that the statement “Cytokine expression increased at 3 hours” is not specific, given this is not a statistical finding. We have removed that sentence. We have also made other slight revisions to clarify the results. The calculation of percent change is now described in the methods at the end of the Multiplex Immunoassay to Measure Functional Responses to PBMCs paragraph. This has also been added to the supplemental figure legend.

The interpretation of results is still insufficient, failing to convey the significance of the findings. 

A paragraph has been added to the discussion to highlight the significance of the findings as well as to highlight to potential for future studies within this population.

Units for alcohol exposure (ounces of absolute alcohol per day) in Table 1 and units for definition as moderate in the PAE group (less than 7 drinks per week) are not consistent. A clearer report of the calculation and assessment is needed.

We have clarified the information presented in Table 1 (see comment above) and have added the following to the definition of moderate PAE group: less than 7 drinks per week, or less than 0.5 AAD (ounces of absolute alcohol per day).